# Conversion of Cannabidiol (CBD) into Psychotropic Cannabinoids Including Tetrahydrocannabinol (THC): A Controversy in the Scientific Literature

**DOI:** 10.3390/toxics8020041

**Published:** 2020-06-03

**Authors:** Patricia Golombek, Marco Müller, Ines Barthlott, Constanze Sproll, Dirk W. Lachenmeier

**Affiliations:** Chemisches und Veterinäruntersuchungsamt (CVUA) Karlsruhe, Weissenburger Straße 3, 76187 Karlsruhe, Germany; golombek.patricia@gmail.com (P.G.); marcomueller17@t-online.de (M.M.); ines.barthlott@student.kit.edu (I.B.); constanze.sproll@cvuaka.bwl.de (C.S.)

**Keywords:** cannabidiol, tetrahydrocannabinol, degradation, psychotropic effects, *Cannabis sativa*

## Abstract

Cannabidiol (CBD) is a naturally occurring, non-psychotropic cannabinoid of the hemp plant *Cannabis sativa* L. and has been known to induce several physiological and pharmacological effects. While CBD is approved as a medicinal product subject to prescription, it is also widely sold over the counter (OTC) in the form of food supplements, cosmetics and electronic cigarette liquids. However, regulatory difficulties arise from its origin being a narcotic plant or its status as an unapproved novel food ingredient. Regarding the consumer safety of these OTC products, the question whether or not CBD might be degraded into psychotropic cannabinoids, most prominently tetrahydrocannabinol (THC), under in vivo conditions initiated an ongoing scientific debate. This feature review aims to summarize the current knowledge of CBD degradation processes, specifically the results of in vitro and in vivo studies. Additionally, the literature on psychotropic effects of cannabinoids was carefully studied with a focus on the degradants and metabolites of CBD, but data were found to be sparse. While the literature is contradictory, most studies suggest that CBD is not converted to psychotropic THC under in vivo conditions. Nevertheless, it is certain that CBD degrades to psychotropic products in acidic environments. Hence, the storage stability of commercial formulations requires more attention in the future.

## 1. Introduction

The hemp plant *Cannabis sativa* L. naturally contains a number of different cannabinoids that are related to the elementary chemical structure of cannabinol (CBN, Figure 1a) [1]. The most prominent representative among the class of these compounds is Δ^9^-tetrahydrocannabinol (Δ^9^-THC, Figure 1b), which is hydrogenated in positions 6a and 7 [1]. Due to the well-known psychotropic properties of Δ^9^-THC, only the cultivation of plant varieties with low contents of Δ^9^-THC is authorized in the European Union (EU) at the moment [2,3]. There is a discrepancy in terms of the legality of products derived from the hemp plant. In general, cannabis products (including flowering or fruiting tops of *Cannabis sativa*) are listed in the United Nations (UN) single convention on narcotic drugs from 1961 [4] and are therefore prohibited regardless of their Δ^9^-THC content. However, processed products, which contain hemp leaves are often regarded as safe and therefore legal, if the Δ^9^-THC content does not exceed certain levels and an abuse as a narcotic drug can be ruled out. As explicitly excluded by the definition of cannabis in the UN single convention [4], seed products (e.g., hemp seed oil), without the cannabinoid-rich resin, are generally regarded as safe and may be marketed in the EU [5,6].

Besides Δ^9^-THC, the non-psychotropic cannabidiol (CBD, Figure 1c) gained increasing popularity due to a broad spectrum of health-promoting effects ascribed to it with several reviews on safety and efficacy available [7,8,9,10,11,12,13,14,15,16,17,18]. In recent years, this culminated in extensive consumer interest with heavily increasing numbers starting in 2018 (Figure 2). Since then, so-called CBD extracts used as a food constituent, in cosmetic products or in the liquids for electronic cigarettes are found with a large variety in drug stores or in online shops [19,20,21]. According to the Novel Food Regulation (EU) 2015/2283, an approval of CBD extracts for the use in food requires a history of food consumption prior to May 1997 [3,22]. Thus, as such a history has not been demonstrated so far, CBD extracts are classified as Novel Food and are therefore not authorized in the EU.

In addition to the discrepancy between the excessive product availability and the doubtful compliance with legislation (EU and worldwide) for many of those products [24,25], questions also arise in regard to the safety of these products. Regarding this, a major topic, which is discussed controversially in the recent scientific literature, is the potential conversion of CBD into psychotropic cannabinoids, including Δ^9^-THC. The observation that CBD products may still induce some psychotropic effects, with various discussed mechanisms including direct action, degradation during storage or under in vivo conditions, as well as contamination, recently highlighted the importance of this question again [19].

This review aims to provide an overview of contemporary and older publications dealing with the conversion of CBD to other (psychotropic) cannabinoids. After a detailed summary on the psychotropicity of cannabinoids (Section 3.1), a comprehensive overview of the conversion of CBD in different conditions is presented (Section 3.2). To provide a better understanding of the pitfalls of cannabinoid research that may possibly account for some controversial results, some major analytical challenges are presented in Section 3.2.1 before the conversion of CBD under acidic conditions (Section 3.2.2) and in vitro conditions (using artificial or simulated gastric juice) is carefully discussed (Section 3.2.3). The current debate about whether results of these studies and some in vivo studies in animals may be transferred to in vivo conditions in humans is elucidated in Section 3.2.4. Finally, the debate about the in vivo conversion of CBD is expanded by the question of whether CBD may convert to other (potentially psychotropic) cannabinoids under storage conditions (Section 3.2.5), and conclusions regarding the risk assessment of CBD products upon oral consumption are provided.

## 2. Materials and Methods

A database research in April 2020 was conducted in Google Scholar and PubMed using the keywords “conversion and/or degradation and/or isomerization of cannabidiol” as well as “structure-activity relationship cannabinoids”, “receptor binding cannabinoids” and “psychoactivity/psychotropicity cannabinoids”. Other aspects of pharmacological effects of CBD not relevant to the aim of the study were excluded.

## 3. Results and Discussion

### 3.1. Psychotropicity of Cannabinoids

The consumption of distinct parts of *Cannabis sativa* L., leading to psychotropic effects, has been known for thousands of years [26]. There are two prominent products mainly used as a drug to willingly induce states of intoxication, which are called marihuana (dried leaves and flowering tops of the *Cannabis sativa* plant) and hashish (resin extracted from the *Cannabis sativa* plant) [27]. The UN suggested prohibition of cannabis and extracts of cannabis in the single convention on narcotic drugs in 1961 [4].

Pharmacological experiments with mixtures and/or single cannabinoids can be traced back to the 1940s and 1950s, when a number of studies regarding THC, CBN and CBD were published regardless of the fact that chemical structures were only elucidated in the mid-1960s [26,28,29,30]. The known psychotropic effects of cannabis were mainly attributed to Δ^9^-THC as a consequence to substantial research during the mid-1960s and early 1970s [30]. The detailed understanding of the biochemical processes induced by cannabinoids (predominantly Δ^9^-THC) was mainly achieved by the discovery of cannabinoid receptors by Howlett et al. [31,32], which ultimately prompted the discovery of endogenous cannabinoids, among which anandamide is the most prominent one [33].

To discuss their psychotropic effects, the most common cannabinoids may be divided into two major groups according to the number of rings in the molecule. The first group is composed of tricyclic cannabinols including CBN, and all THC and hexahydrocannabinol (HHC, Figure 3a) isomers. The second group—which will be discussed later in this section—consists of bicyclic cannabinoids with CBD, cannabigerol (CBG, Figure 3b) and cannabichromene (CBC, Figure 3c) being the most prominent representatives.

In 1971, the UN released a convention listing psychotropic substances in four schedules ranging from Schedule I (most restrictive) to Schedule IV (least restrictive) [34]. This convention classifies five THC derivatives (i.e., Δ^6a^-THC, Δ^7^-THC, Δ^8^-THC, Δ^10^-THC and Δ^10a^-THC) and HHC in Schedule I, while Δ^9^-THC is listed in Schedule II. However, scientific data on the psychotropic effects of these substances are rather limited. Only recently the Expert Committee on Drug Dependence of the World Health Organization (WHO) released a critical review on isomers of THC, stating that Δ^8^-THC and Δ^9(11)^-THC were found to exhibit Δ^9^-THC like effects when administrated to different animals, whereas Δ^10^-THC lacked a comparable effect [35]. Moreover, Δ^8^-THC and Δ^6a(10a)^-THC were found to have psychotropic effects on humans while data regarding the effect of other THC derivatives on humans are still missing [35].

The well-established fact that THC derivatives are metabolized by means of hydroxylation at C11 prompted Lemberger et al. [36] to investigate 11-hydroxy-Δ^9^-THC and Watanabe et al. [37] to study the psychotropic effects of 11-hydroxy-Δ^8^-THC. Both studies revealed that these substances show even more enhanced effects than their respective non-hydroxylated forms when administrated to humans. The same effect was also found for 11-oxo-Δ^8^-THC [37]. In another study by Järbe et al. [38], two stereoisomers of 7-hydroxy-HHC showed psychotropic effects on rats and pigeons. According to Watanabe et al. [39], 8-hydroxy-*iso*-HHC (9 mg/kg i.v.) produced a significant hypothermia in mice at 15 to 90 min after administration, while 9α-hydroxy-HHC failed to induce this effect. Both caused a significantly prolonged pentobarbital-induced sleeping (1.8 to 8 times). In summary, both hydroxy-HHCs showed THC-like effects in mice but they were less active than Δ^9^-THC [39].

In contrast to that, the acid forms Δ^8^-tetrahydrocannabinolic acid (Δ^8^-THCA) and Δ^9^-THCA as well as the metabolite 11-COOH-THC failed to cause any observable physiological effect or psychotropic effect, even though detailed studies are missing [40]. However, as mentioned by Moreno-Sanz et al. [41], Δ^9^-THCA slowly decarboxylates to form THC during storage and fermentation but also during the baking of edibles, smoking or vaporizing and may thus exhibit psychotropic effects upon respective consumption.

Despite some early uncertainty regarding the psychotropic effect of CBN as described by Yamamoto et al. [42], Järbe et al. [43] reported on the psychotropic effects of CBN in rats and pigeons. But high doses of up to 14 mg/kg were required, whereas Δ^9^-THC induced similar effects with doses of 3 mg/kg. As also found for other cannabinoids, the hydroxylated form 11-hydroxy-CBN showed more pronounced effects than the non-hydroxylated form [42].

Some general observations were reported by Compton et al. [44], who investigated the correlation of binding affinity with psychotropic effects in humans for various different cannabinoids in a detailed study and found a strong correlation. Interestingly, this study revealed that cannabinoids with long (branched) side chains at C3 do have larger binding affinities compared with ones with short (unbranched) side chains. Besides that, hydroxy groups or halogens located at the terminal end of the side chain (i.e., at C5′ position) induced good binding affinities. Carboxylic acid metabolites of either Δ^9^-THC or Δ^8^-THC, though, were not found to bind to the receptor at all. This is well in agreement with findings of their non-psychotropic effects. The authors additionally reported on different binding characteristics for CBD and Δ^9^-THC and used this as a hypothesis for their different physiological effects [44].

In fact, CBD is described as “non-psychotropic” [45] or even “anti-psychotropic” [11,12] as it does not show effects comparable to Δ^9^-THC, neither in studies on animals as already reported by Mechoulam et al. [46] in 1970 nor in humans as reviewed by Iseger et al. [12]. However, a multitude of psychological and physiological effects (some examples are anti-inflammatory, antiemetic, antipsychotic, anticarcinogenic, anxiolytic and analgesic effects, effects on appetite, positive effects on multiple sclerosis and spinal cord, as well as on Gilles de la Tourette’s syndrome, epilepsy, glaucoma, diabetes, Parkinson disease and dystonia) were associated with CBD and reviewed in a number of articles [7,8,9,10,11,12,13,14,15,16,17,18]. In agreement with the hypothesis by Compton et al. [44], a physiological explanation for the different pharmacology was presented by Pertwee et al. [47], when they reported on the unexpectedly high potency of CBD to act as the antagonist of CB1/CB2 receptors in cells or tissues expressing these receptors. This is in contrast to Δ^9^-THC, which was described as an agonist of the respective receptors [47].

Similar to THC- and HHC-type cannabinoids, the acid form (either methylated or not) of CBD—which is most likely also a product formed during the metabolism of CBD—was found to show some effects on, e.g., cancer and hyperalgesia [48,49], neither of which, though, may be termed psychotropic. Similarly, CBD monomethyl ether (CBD-ME) was found to lack psychotropic activity in a study conducted on rats and pigeons [38].

In line with structural prerequisites, also CBG was termed “non-psychotropic” and, according to binding studies on the cannabinoid receptors CB1 and CB2, may show some beneficial actions and thus exert therapeutic potential such as protection against oxidative stress in macrophages [50]. Even though data on metabolites of CBG is sparse, 5-acetyl-4-hydroxy-CBG was found to have antileishmanial effects [51].

In an excellent review about the chemistry, synthesis and bioactivity of CBC, Pollastro et al. [52] summarized recent studies on the psychotropicity of CBC. Even though no narcotic effect was found in in vivo experiments, high doses of CBC may indeed exhibit responses typical for Δ^9^-THC (e.g., hypomotility, catalepsy, hypothermia and analgesia). The authors claim that the reason for this effect most likely derives from another than the typical mechanism, as CBC was found to show only marginal affinity for the cannabinoid receptors CB1 and CB2 [52]. Besides that, multiple other effects are related to CBC, among which the antibacterial and antifungal activity is the most noteworthy one as CBC outperforms other cannabinoids in this category. According to the authors, little information exists regarding the biological profile of naturally occurring analogs of CBC [52].

Even though research on physiological, psychoactive and psychotropic effects on various cannabinoids has been highly productive in the last 50 years, detailed clinical data of many isomers and metabolites of cannabinoids are still missing. This gets even more relevant in the light of the ongoing debate about the conversion of CBD to several of these compounds (Section 3.2). Further studies on the psychotropicity of the respective conversion products may contribute and help to further clarify the scientific debate about the in vivo activities of CBD.

### 3.2. Conversion of Cannabidiol

Various conversion routes for CBD are reported in the literature. An overview of the broad range of conditions of these reactions and the resulting conversion products is presented in Figure 4 and further discussed in the subsequent chapters. In brief, most of the reactions require acidic conditions, high temperatures or are observed in vitro. Under these conditions, CBD is converted to ∆^9^-THC as well as ∆^7^-THC, ∆^8^-THC, ∆^10^-THC, ∆^11^-THC and *iso*-THC [53,54]. A formation of the hydroxy derivatives 11-hydroxy-CBD, 11-hydroxy-THC, 5′-hydroxy-CBD, 11, 5′-dihydroxy-CBD and 11,5′-dihydroxy-THC was previously reported [54]. Furthermore, a formation of the two HHC derivatives 9α-hydroxy-HHC and 8-hydroxy-*iso*-HHC has been reported [39,54]. In the presence of methanol or ethanol, the methoxy or ethoxy derivatives 9-methoxy-HHC and 10-methoxy-HHC or 9-ethoxy-HHC and 10-ethoxy-HHC are formed [53,54,55]. Besides that, CBN was reported to be formed under in vitro conditions [39]. However, only one of the reported reactions (i.e., the conversion of CBD to Δ^9^-THC) was observed in vivo in rats [56]. A detailed summary of all conversion reactions is also presented in Table A1 (Appendix A).

#### 3.2.1. Analytical Challenges in Detecting CBD, Its Degradation Products and Other Cannabinoids

By the time of their first detection, cannabinoids were mainly analyzed by color reactions such as the Duquénois–Negm test and the Beam test as summarized by Vollner et al. [57]. Some of these tests were highly sensitive and enabled the differentiation between various cannabinoids [57]. Besides that, thin layer chromatography (TLC), photometric and spectroscopic methods were reported as well [57]. The development of gas chromatography (GC) by Martin and Synge in the early 1950s and its immediate success in analytical chemistry [58] soon also reached the field of cannabis research when Farmillo and Davis developed the first GC method to separate a number of different cannabinoids in 1960 [59,60]. Similar to GC, the invention and rise of the high-performance liquid chromatography (HPLC) technique in the late 1960s [61] quite immediately paved its way to the field of cannabis research. First reports on the use of HPLC to detect and quantify several cannabinoids can be ascribed to the working group of R.N. Smith, according to a series of publications starting in 1975 [62,63,64]. Both GC and HPLC are still used as major tools in the analysis of cannabinoids nowadays, yet in more sophisticated versions. Countless reports on MS and MS/MS hyphenation techniques as well as two dimensional approaches (e.g., GC × GC) were reported and reviewed carefully [65,66].

The most important drawback of GC was already reported by Farmillo and Davis in one of their first publications [60]. Due to high temperatures in the injector port and the column oven, acidic forms of cannabinoids are decarboxylated and are thus not detected in the resulting chromatogram. While this causes an underestimation of such compounds, it may also lead to a substantial overestimation of the decarboxylated forms [66]. However, according to Dussy et al. [67], thermal conversion reactions under typical GC conditions are not limited to decarboxylation processes and expressing the calculated amount of the decarboxylated form as a sum of the acidic and decarboxylated form leads to a certain underestimation.

Moreover, thermal reactions possibly also occur for other derivatives of cannabinoids, such as hydroxylated or methoxylated forms, which complicates the use of GC in this field. Notably, first hints for thermal conversion of CBC to Δ^9^-THC (both with their pentyl side chain substituted by hydrogen atoms) were reported by Garcia et al. [68]. This could most likely also apply to CBD and Δ^9^-THC. In regard to this, an interesting observation was that all in vivo studies, which detected Δ^9^-THC after the administration of CBD [56,69,70], applied GC/MS methods, while other studies were conducted using liquid chromatography (LC)/MS or LC-MS/MS methods. Hence, the question arises whether Δ^9^-THC may not be formed in vivo but rather artifactually based on thermal reactions in the GC/MS system. Notably, efforts by applying derivatization, mainly by using trimethylsilylation with N,O-Bis(trimethylsilyl)trifluoroacetamide (BSTFA) and chlorotrimethylsilane (TMCS) [65] tremendously improved GC separations and even enabled the detection of acidic forms. However, the derivatization process was sometimes found to be not quantitative [66,71]. Hence, it cannot be excluded that even small amounts of CBD which evaded derivatization may account for positive findings of Δ^9^-THC due to thermal reactions in the GC system.

However, not only results obtained by GC/MS need to be considered with a certain amount of scrutiny. As multiple cannabinoids (e.g., Δ^9^-THC, CBD and CBC) are isobaric isomers, they form identical signals and mass spectra even with LC-MS/MS measurements [19]. Hence, chromatographical methods with high separation performance are required for an unambiguous peak assignment and avoidance of false positive results. Interestingly, Broecker et al. [72] also reported on the acid-catalyzed in-source equilibration of Δ^9^-THC and CBD after positive electrospray ionization (ESI) in flow-injection experiments. Careful studies with H/D exchange experiments proved that CBD and Δ^9^-THC may not be distinguishable by mass spectra alone. The authors thus highlighted the great importance of using the retention time in LC-MS/MS measurements to distinguish different cannabinoids. Problems arise for small abundant compounds, such as isomers or degradation products, which could very likely exhibit similar retention times to other more abundant compounds. For example, Kiselak et al. [54] claim in their manuscript that “LC/MS analysis was able to separate all of the psychotropic cannabinoids”. However, according to the chromatograms shown in Figure 2, Figure 3, Figure 4 and Figure 5 of the respective manuscript, the LC separation shows multiple peaks and shoulder peaks under the best separation conditions and may therefore not be complete for some isobaric isomers besides Δ^8^- and Δ^9^-THC and coelutions cannot be excluded.

This is further complicated by the fact that not all isomers are distinguishable due to similar MS/MS fragmentation patterns. Such problems were recently reported by Lachenmeier et al. [19], as they identified minor compounds with MS/MS fragmentation patterns similar to CBN and Δ^9^-THC but were unable to structurally assign them. In the discussion of their results, the authors draw attention to the problem that data obtained with less selective and specific chromatographical methods might easily lead to a mix-up of certain CBD degradation products with THC isomers, besides Δ^8^- and Δ^9^-THC, due to structural similarities accompanied with nearly identical retention times. According to the authors, this can account as a possible explanation for the (potentially false positive) detection of Δ^9^-THC in some previous studies.

Despite the various problems arising from analytical challenges in the field of cannabis research, only 2% of all publications on cannabis deal with analytics, as stated by Gertsch et al. [73] in the editorial of a recently published special issue on cannabis. Hence, when comparing results of in vitro (Section 3.2.3) or in vivo (Section 3.2.4) studies on the conversion of CBD, analytical challenges need to be considered and all claims should be critically assessed.

#### 3.2.2. Conversion of CBD under Acidic Conditions

The acid-catalyzed conversion of CBD has been studied since the early 1940s, when Adams et al. [74] reported on the treatment of CBD with various acids. While adding trichloroacetic acid, anhydrous oxalic acid, picric acid, 3,5-dinitrobenzoic acid, 87% formic acid, glacial acetic acid and malic acid to a solution of CBD in benzene did not result in conversion even after boiling for 10–20 h, good results were achieved with the addition of dilute ethanolic hydrochloric acid, *p*-toluenesulfonic acid or a drop of sulfuric acid (100%) in cyclohexane. The conversion product was described to be a psychotropic cannabinoid, which the authors assumed to be either Δ^9^-THC or Δ^8^-THC (Figure 4). This observation was later confirmed by Gaoni and Mechoulam [55], who described the correct structures of CBD, Δ^9^-THC and Δ^8^-THC based on careful spectroscopic studies (i.e., UV, IR and NMR measurements). They were further able to verify the hypothesis of Adams et al. [75] that Δ^9^-THC was the main product if CBD was subjected to treatment with hydrochloric acid. The addition of *p*-toluenesulfonic acid, though, rather resulted in the formation of Δ^8^-THC. 

Higher product yields of either Δ^8^-THC or Δ^9^-THC can be gained by means of the improved conditions presented by Webster et al. [76]. The conversion of CBD to Δ^8^-THC is enhanced, if a CBD solution in toluene is boiled in the presence of a Lewis acid (*p*-toluenesulfonic acid or boron trifluoride, BF_3_), while Δ^9^-THC is preferably formed when CBD is dissolved in dichloromethane (DCM) and stirred at 0 °C in the presence of boron trifluoride etherate (BF_3_Et_2_O). To avoid the formation of oxidized side products, Webster et al. [76] further recommended conducting the conversion of CBD to Δ^8^-THC or Δ^9^-THC under nitrogen atmosphere. This leads to the question which additional products are formed in the presence of oxygen or oxidative agents. In a series of publications, Gaoni and Mechoulam [53,55,77] reported on the formation of methoxy/ethoxy HHCs (Figure 4) upon boiling a CBD solution in methanol/ethanol for 18 h in the presence of diluted sulfuric acid or hydrochloric acid (HCl). A methoxy or ethoxy group was introduced either in the 9- or 10-position, resulting in two distinct isomers, which were 9-ethoxy/methoxy-HHC and 10-ethoxy/methoxy-HHC, respectively (Figure 4). Besides the above-mentioned products, the reaction mixture also contained Δ^9^-THC, Δ^8^-THC and *iso*-THC (structure in Figure 4). The latter was also found with a yield of 13% when a solution of CBD in DCM/chloroform was boiled in the presence of BF_3_Et_2_O (Figure 4).

Layton et al. [78] investigated possible formation products when crystalline CBD was treated with 3% hydrogen peroxide, 0.1 M sodium hydroxide (NaOH) or 0.1 M HCl. While oxidative and basic conditions produced little to no conversion products, acidic conditions resulted in the formation of Δ^9^-THC and Δ^8^-THC besides another cannabinoid, which showed the same ultra-performance liquid chromatography (UPLC)-MS retention time as CBG. However, according to a high signal at *m/z* 333 in the mass spectrum of the third compound, CBG (typical fragment *m/z* 317) was ruled out as a possible formation product. In the light of recent studies by Kiselak et al. [54], the unknown compound may tentatively be assigned to a hydroxy form of either CBD or THC.

In this study, Kiselak et al. [54] also reported on the conversion of CBD dissolved in ethanol and refluxed for 24 h in the presence of battery acid (sulfuric acid), muriatic acid (HCl) or vinegar (acetic acid). While sulfuric acid resulted in a full turnover of CBD after 4 h, the other two acids did not lead to a complete isomerization of CBD even after 24 h. Careful studies by means of ion mobility-coupled LC-MS/MS measurements enabled the detection of various formation products. Besides Δ^9^-THC, the products 8-hydroxy-*iso*-HHC, 11-hydroxy-THC, 11,5′-dihydroxy-CBD, 11,5′-dihydroxy-Δ^9^-THC, 11-hydroxy-CBD, 9α-hydroxy-HHC, 5′-hydroxy-CBD, Δ^7^-THC, Δ^8^-THC, Δ^10^-THC, Δ^11^-THC, 9-methoxy-THC and 10-methoxy-THC were identified (Figure 4). Peak identification was accomplished by comparison with the retention times of the reference standards and structures of unknown peaks were assigned using data from the MS/MS fragmentation and ion mobility. Yet, the only available reference substances were Δ^8^-THC, Δ^9^-THC, CBD, CBG, CBN, THCA (Figure 5a), cannabidiolic acid (CBDA, Figure 5b) and CBC. The product pattern varied depending on the reaction conditions. HCl yielded the largest number of products and exclusively led to the formation of 11,5′-dihydroxy-Δ^9^-THC. The reaction with sulfuric acid was the only one to produce 10-methoxy-THC and the addition of acetic acid was the only method to produce 5′-hydroxy-CBD. Interestingly, 11-hydroxy-CBD was formed in all reactions [54].

In the light of the reported results on the broad spectrum of products from the acid-catalyzed conversions of CBD, the question arises if similar reactions are also found in the acidic conditions of (artificial) gastric juice (Section 3.2.3).

#### 3.2.3. In Vitro Studies: Conversion of CBD in Artificial Gastric Juice and Other Model Systems

Despite the importance to understand the conversion reactions of CBD in the presence of animal or human cells or enzymes, the number of published studies reporting on in vitro studies of CBD is rather small. The first report on the biotransformation of CBD to a derivative of the psychotropic Δ^9^-THC was presented by Nagai et al. in 1993 [79]. In their experiments, the authors incubated a CBD solution with hepatic microsomes of guinea pigs, rats and mice, extracted the mixture with ethyl acetate, analyzed the resulting extract with GC/MS and identified 6β-hydroxymethyl-Δ^9^-THC. 

Fourteen years later, the studies were continued by Watanabe et al. [39], who found that CBD was converted to 9α-hydroxy-HHC, 8-hydroxy-*iso*-HHC, Δ^9^-THC and CBN when subjected with artificial gastric juice (without pepsin) and incubated at 37 °C for 20 h. The analysis of the ethyl acetate extracts was carried out by GC/MS and structures were assigned by mass spectral data and retention times as compared to the self-synthesized (or isolated) reference standards (Δ^9^-THC, CBD, CBN, 8-hydroxy-*iso*-HHC and 9α-hydroxy-HHC).

More recently, Merrick et al. [80] investigated conversion products of CBD, which were formed upon subjection with simulated gastric juice and a physiological buffer solution of 4-(2-hydroxyethyl)-1-piperazineethanesulfonic acid (HEPES). Both solutions additionally contained 1% sodium dodecyl sulfate (SDS) to improve solubility, as recommended by United States Pharmacopeia (USP). Based on UPLC/UV and UPLC-MS/MS analyses, Δ^9^-THC and Δ^8^-THC were detected in simulated gastric juice and HEPES buffer after one to three hours of incubation. This led the authors to the conclusion that relevant levels of Δ^9^-THC and Δ^8^-THC may be formed in humans after oral consumption of CBD. This statement was criticized by other scientists as animal and human clinical studies did not provide evidence for the conversion of CBD to THC in vivo (see Section 3.2.4) [81].

In contradiction to the above-mentioned results, another recent study conducted by Lachenmeier et al. [19] reported no observation of the formation of THC (neither the Δ^9^- nor the Δ^8^-form) when CBD dissolved in methanol was incubated with artificial gastric juice or stored under stress factors such as heat or light under moderate conditions. Only if a solution of CBD in methanol was acidified with 0.5 mol/L HCl and stored for up to two weeks, a complete degradation of CBD and formation of 27% THC were reported. This study attached great importance to the physiological study design, especially in regard to incubation times, temperatures during incubation and concentrations of solvents and analytes. In the controversial debate about whether or not Δ^9^-THC is formed under in vitro conditions (i.e., with simulated gastric juice) and—based on that—also in in vivo conditions (reviewed in Section 3.2.4), the authors thus positioned themselves on the opposing side. 

#### 3.2.4. In Vivo Studies: Conversion of CBD in Animals and Humans

Even though the metabolism of CBD was studied in several animal species (e.g., dogs and rats) [82,83,84,85] before, Harvey and Mechoulam were the first to report on the human CBD metabolism in the early 1990s [69,70]. As they measured human urine samples with a GC/MS method after the patients were orally administrated with CBD, they found over 30 metabolites, which were mainly hydroxylated in various positions. Interestingly, Harvey and Mechoulam also reported on the detection of two cyclized cannabinoids, which they termed “delta-6-THC” and “delta-1-THC” (the latter one most likely corresponds to Δ^9^-THC as termed by present nomenclature) [69]. They concluded that these analytes rather emerged artifactually in the urine sample than being metabolites formed in humans, as this would have caused “psychoactivity with obvious adverse effects for the patient” [69].

This hypothesis was supported by findings of Consroe et al. [86], who treated 14 patients with Huntington disease with a CBD dose of 10 mg/kg/day and compared plasma CBD levels with a group treated with a placebo. Over the course of six weeks, Δ^9^-THC was not detected in the plasma. Similar results were also reported by Martin-Santos et al. [87] while conducting a double-blinded study with 16 healthy volunteers treated with either Δ^9^-THC (10 mg), CBD (600 mg) or a placebo. Neither Δ^9^-THC, 11-hydroxy-THC or 11-nor-9-carboxy-tetrahydrocannabinol (11-COOH-THC) were detected in significant amounts in the blood of patients treated with CBD, while the oral administration of Δ^9^-THC itself had both effects on the plasma concentration and measurable psychotropicity. In a recent review article, Ujváry et al. [88] further summarized literature data on CBD metabolites and human metabolic pathways of CBD.

More recently, the question whether or not CBD may be converted to Δ^9^-THC or to other (potentially) psychotropic cannabinoids under in vivo conditions after oral administration culminated in a scientific debate, which was mainly initiated by an article published by Merrick et al. [80], who studied the in vitro conversion of CBD to Δ^9^-THC and concluded that this can be applied to in vivo situations as well. In a direct rebuttal to this publication, Grotenhermen et al. [81] cited multiple clinical studies on CBD administration to human volunteers, which rule out psychotropic effects of CBD and thus a conversion to Δ^9^-THC. In reply to Grotenhermen’s rebuttal letter, Bonn-Miller et al. [89] stressed the multiple recent studies that proved the conversion of CBD to Δ^9^-THC in an acidic environment (such as simulated gastric juice) and indicated the lack of data and the need for further human clinical studies. These studies should also monitor the formation of other cannabinoids, such as 9α-hydroxy-HHC or 8-hydroxy-*iso*-HHC. In an immediate response, Nahler et al. [90] argued that a conversion of CBD to Δ^9^-THC does not occur in humans and stated that simulated gastric juice might not sufficiently reflect conditions in the human body. According to the authors, if Δ^9^-THC was formed upon oral administration of CBD in the human stomach, Δ^9^-THC and its metabolites 11-hydroxy-THC and 11-COOH-THC should be detectable in the serum as well. However, Nahler et al. [90] found no evidence for that when analyzing previously published results.

In the timeframe of only one year following this debate, several articles were published that either support one or the other side. For example, Palazzoli et al. [91] did neither detect Δ^9^-THC nor the metabolites 11-hydroxy-THC or 11-COOH-THC (or its glucuronides) in the whole blood of male rats 3 or 6 h after an oral CBD dose of 50 mg/kg in olive oil. Notably, this study was conducted with an LC–MS/MS method. A similar method was also used when Wrey et al. [92] examined blood/plasma samples of minipigs after they were given a dose of 15 mg/kg CBD in sesame oil (twice a day, for four days with a single final dose at day five). Similar to Palazzoli et al. [91], Wrey et al. [92] did not detect Δ^9^-THC or one of its metabolites 1, 2, 4 or 6 h after oral administration of CBD to the minipigs.

In contradiction to this, Hložek et al. [56] were able to detect Δ^9^-THC in the serum and in the brain of rats after they were administrated with doses of 60 mg/kg CBD. Even lower doses of 10 mg/kg CBD caused Δ^9^-THC to be detected in the serum, while it was not detectable in the brain. According to the authors, Δ^9^-THC levels in the brain may have been below the limit of detection of their GC/MS method. The authors further stated that these findings remain to be demonstrated in humans. Only recently, Crippa et al. [93] published an article about a pharmacokinetic study in 120 healthy human subjects. They found that orally administered CBD (300 mg as corn oil formulation) was not converted to Δ^8^-THC or Δ^9^-THC in humans. None of the different THC forms were detected in the whole blood 3 and 6 h after intake by means of an LC-MS/MS method.

As a side note to the ongoing debate, it should be mentioned that using highly pure CBD is of utmost importance. Crude extracts, which contain other cannabinoids, such as Δ^9^-THC, may not only cause false results but may also lead to psychotropic effects in humans [19,94]. The use of a non-pure CBD reference material (e.g., due to conversion during storage (Section 3.2.5) or an insufficient isolation process) may explain the controversial findings by Hložek et al. [56] on the one side and Palazzoli et al. [91] or Wrey et al. [92] on the other side. Different studies may also be distinguished by the analytical method (GC/MS or LC-MS/MS) used for the detection and/or quantification of cannabinoids. Especially with regard to the problems related with either of these techniques, it seems to be useful to critically asses this aspect (Section 3.2.1). 

Finally, recent studies mainly focused on the conversion of CBD to Δ^9^-THC and its metabolites. While this is—beyond doubt—currently the most important psychotropic cannabinoid, the conversion of CBD may potentially also lead to other products that can cause psychotropic effects (Section 3.1), which were not examined by most of the studies so far. Despite the increasing number of publications driven by the ongoing scientific controversy, the question if conversion processes of CBD may lead to psychotropic effects in the human body is still not answered conclusively.

#### 3.2.5. Conversion of CBD during the Storage of CBD Products

Due to the heavily increasing trend of CBD products on the market (Figure 2), the consumer safety in regard to these products is of great interest. In a recent publication by Lachenmeier et al. [21], the authors listed multiple CBD products, which contained significant amounts of Δ^9^-THC and were thus reported in the Rapid Alert System for Food and Feed (RASFF) of the EU. In another publication, Lachenmeier et al. [19] reported on consumer complaints noting “THC-like effects” after consumption of CBD products. The authors discussed three hypotheses for this effect, of which the first one that CBD may have a psychotropic action itself, was immediately ruled out due to missing scientific evidence that CBD exhibits psychotropic effects as compared to Δ^9^-THC. Another reason for the psychotropic effects may be explained by the transformation of CBD to Δ^9^-THC under in vivo conditions. Even though the authors rather neglected that option due to the results of their own conversion studies, the scientific debate about this hypothesis is still ongoing (Section 3.2.4). A third reason discussed by the authors is that Δ^9^-THC may already be present in the CBD products as contamination, e.g., due to the use of crude hemp extracts instead of purified CBD. This point was also highlighted in a recent investigation of Liebling et al. [95], as they found multiple cannabinoids (including psychotropic forms) in over-the-counter CBD products in the UK.

A further hypothesis, which was not discussed in the mentioned article by Lachenmeier et al. [19], is that other psychotropic cannabinoids (e.g., Δ^9^-THC) are not present in the original CBD extract or CBD product, but potentially result from chemical reactions under storage conditions. The most obvious and indisputable case is the storage of CBD products under acidic conditions, which facilitates the conversion of CBD into Δ^9^-THC, as proven by many studies (Section 3.2.2). Such conditions are sometimes found in liquids for electronic cigarettes resulting in the need of special attention for these products. Besides the well-studied effects of acidic conditions, also basic conditions were found to lead to conversion products of CBD. As reported by Srebnik et al. [96], CBD isomerizes in a high yield to Δ^6^-CBD upon heating with *t*-pentyl potassium in toluene-hexamethyl-phosphoric triamide (6:1, v/v). This is especially interesting as Δ^6^-CBD exhibits THC-like effects in rhesus monkeys according to Mechoulam et al. [97]. Additionally, several recent investigations [20,24,98,99,100] demonstrated that CBD products often contain much less CBD than declared. This leads to the question if other than acidic (or basic) conditions may contribute to the conversion of CBD into further (potentially psychotropic) cannabinoids as well.

During the 1970s, the decomposition of CBD in various solvents was controversially discussed between Turner et al. [101], who reported no decomposition, and Fairbairn et al. [102] as well as Parker et al. [103], who indeed found CBD to be decomposed in different solvents under storage conditions. When Smith et al. [104] stored different cannabinoids at -18 °C in darkness, they only found small decomposition rates, which slightly increased at room temperature (20 °C), depending on the cannabinoid (decomposition rate of Δ^9^-THC higher than that of CBD). However, light seemed to spark the decomposition as the rates were significantly higher in daylight conditions. Unfortunately though, no conversion products were measured in this study [104]. Another study was conducted by Lyndon et al. [105], who found that CBD is decomposed by 11% upon UV irradiation for seven days. As this decay was not associated with an increase in Δ^9^-THC, the authors concluded that photochemical conversions of CBD to Δ^9^-THC “probably do not occur”. As Δ^9^-THC is decomposed itself, a closer look into the decomposition rates appears necessary.

The mechanism of the main decomposition route for Δ^9^-THC, which ultimately leads to the formation of CBN, was reported by Turner et al. [106]. This was proved by Harvey et al. [107], when they found low levels of Δ^9^-THC but increased levels of CBN in marihuana samples stored for nearly 100 years. Further, Lindholst et al. [108] identified CBN as the resulting product from Δ^9^-THC decomposition in a study on cannabis resin stored over the course of four years. Another long-term study by Trofin et al. [109,110] reported on the decomposition of Δ^9^-THC but also CBD to the final product CBN in samples stored for four years in different conditions. As the decay of CBD was half the difference between the decay of Δ^9^-THC and the formation of CBN, the authors postulated the degradation route of CBD to start with a cyclization to Δ^9^-THC, which is followed by the decomposition to CBN. Notably, room temperature and daylight were found to increase the decomposition rates in the studies of Trofin et al. [109,110] and Lindholst et al. [108]. An interesting finding was reported by Skopp et al. [111], who claimed that CBN might not be the final conversion product in keratinized hair samples, as it could be further degraded by a light-induced radical reaction.

A recent report by Grafström et al. [112] also highlights the role of oxygen in the decomposition process, as samples stored in contact with air showed higher decomposition rates of Δ^9^-THC and CBD both in daylight and dark conditions. The authors additionally reported on a greater stability of CBD as compared with Δ^9^-THC, regardless of the applied conditions. However, Δ^8^-*iso*-THC detected in the samples was reported to arise from a ring closure between the phenolic OH group and the endo double bond within the CBD molecule. Acidic forms of cannabinoids (e.g., Δ^9^-THCA and CBDA) are more prone to degradation than their respective non-acid forms and notable decay was also found in dark conditions with higher rates at room temperature than at 4 °C or −20 °C [108].

Hence, next to the effects in acidic conditions, first hints on the effects in basic conditions as well as the reported decomposition processes and their dependence on temperature, light and available oxygen need to be considered when storing CBD products. The described cyclization of CBD to Δ^9^-THC may lead to psychotropic effects and to potential harm for the consumer of the respective product. It has to be mentioned, though, that most of the findings were reported for long-term storage tests with time frames considerably exceeding typical storage times of CBD products. Moreover, as the decomposition rate of Δ^9^-THC was reported to be higher than the one of CBD, large amounts of Δ^9^-THC resulting from decomposition processes are not to be expected in stored products. Regardless of that, the accumulation of CBN formed in decomposition processes during long-term storage should be avoided due to potential psychotropic effects related to CBN. Thus, the storage of CBD products needs to be carefully monitored.

## 4. Conclusions

The increasing number of publications related to the pharmacological effects of CBD has stimulated marketers of CBD products to advertise their goods with specific health claims, despite a lack of clinical evidence in most cases [113]. Along with the increasing number of such products on the market, this opens up concerns regarding consumer safety and consumer deception related to the efficacy of these articles. One of these questions is the potential conversion of CBD to psychotropic cannabinoids under in vitro and in vivo conditions, which is currently the topic of an ongoing scientific debate. A conversion of CBD to the psychotropic forms Δ^9^-THC and Δ^8^-THC upon treatment with strong acids, such as hydrochloric acid, sulfuric acid or *p*-toluenesulfonic acid, was doubtlessly proved by many publications. Some of these findings were demonstrated to also occur under in vitro conditions, e.g., by using artificial gastric juice for incubation.

The transfer of these results to in vivo conditions seems to be the major point of the ongoing controversy as the in vivo conversion of CBD to Δ^9^-THC was not supported by the majority of the animal studies, where neither Δ^9^-THC nor one of its metabolites 11-hydroxy-THC and 11-COOH-THCA were detected in blood or in brain tissues. Adding to this, neither Δ^9^-THC nor any of its metabolites were detected after oral CBD administration in any of the human studies. Difficulties arising from detection methods such as GC/MS and LC-MS/MS may help to explain some of the contradictory results, contributing to the ongoing debate. Nevertheless, most of the published data support the conclusion that upon oral consumption of CBD products, a conversion of CBD to an amount of Δ^9^-THC that exceeds the threshold of pharmacological action is not very likely in the human organism. 

A comprehensive risk assessment of CBD products, however, not only requires the monitoring of an in vivo formation of Δ^9^-THC (or other psychotropic cannabinoids) but also the pre-consumption reactions occurring in the product itself. The strongest and the most clinically relevant piece of evidence determined in this review in favor of CBD’s conversion to psychotropic metabolites is during improper storage. For example, CBD may cyclize to Δ^9^-THC under storage conditions, even though both compounds are further degraded to CBN, which in turn may exhibit psychotropic effects itself. Hence, there is a special need for manufacturers to include shelf-life studies dedicated to the long-term stability of CBD in the finished products, considering the formation of psychotropic compounds by the degradation of CBD. Accordingly, an interesting possibility would also be testing for compounds or conditions that help to prevent or slow down CBD degradation, comparable to antioxidants used to protect lipid compounds in food from oxidation.

## Figures and Tables

**Figure 1 toxics-08-00041-f001:**
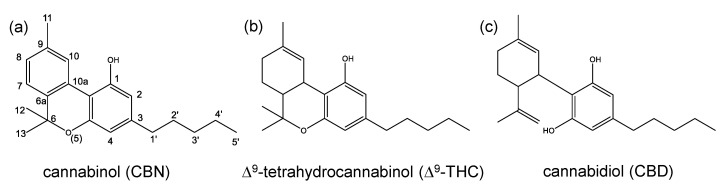
Chemical structures of (**a**) cannabinol (CBN) including the numbering system, (**b**) Δ^9^-tetrahydrocannabinol (Δ^9^-THC) and (**c**) cannabidiol (CBD).

**Figure 2 toxics-08-00041-f002:**
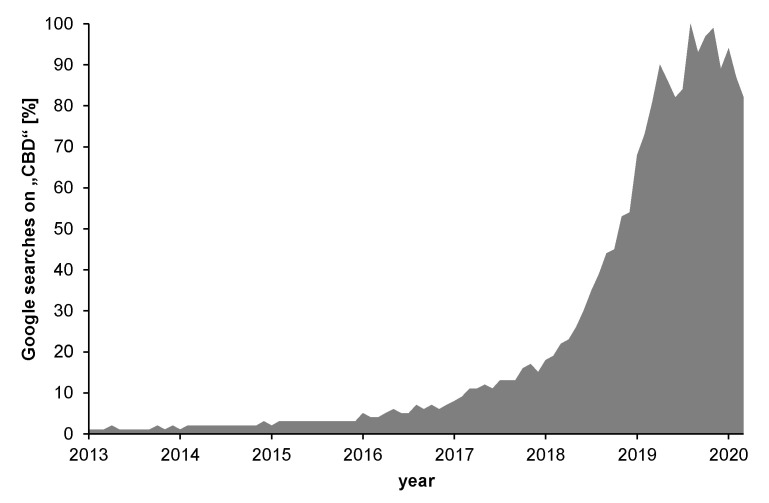
Google trends analysis for cannabidiol (CBD) (Data source: Google Trends [23]).

**Figure 3 toxics-08-00041-f003:**
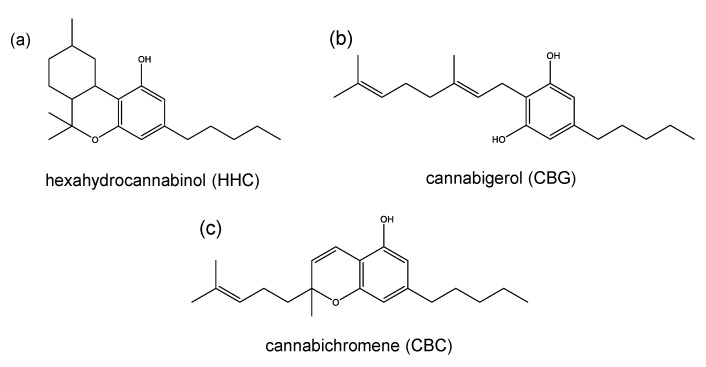
Chemical structures of (**a**) hexahydrocannabinol (HHC), (**b**) cannabigerol (CBG) and (**c**) cannabichromene (CBC).

**Figure 4 toxics-08-00041-f004:**
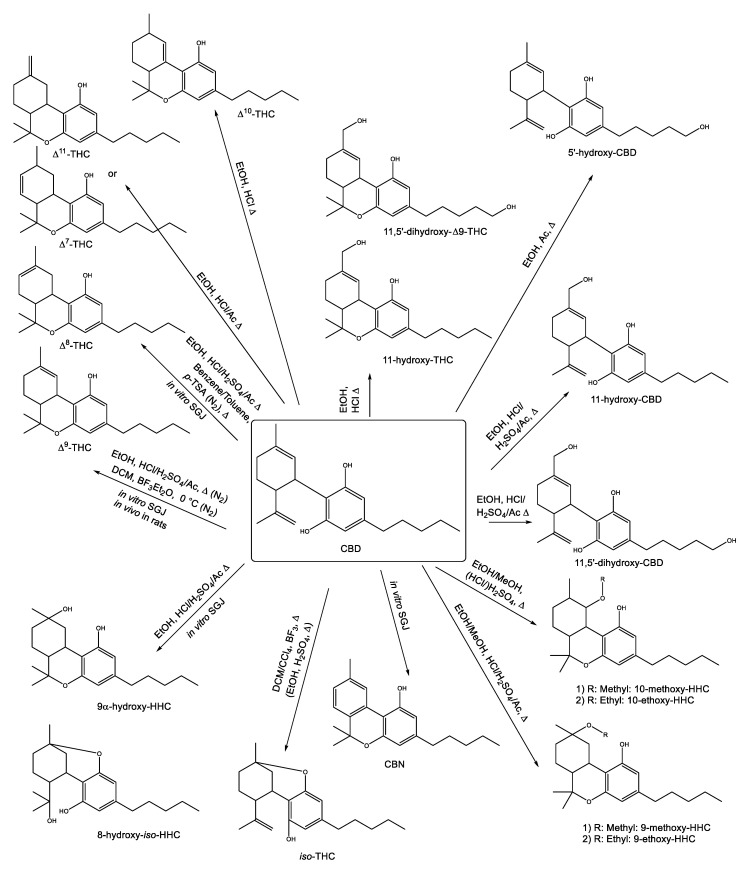
Overview of various chemical conversions of cannabidiol (CBD) to different conversion products and the respective conditions, which are reported in the literature.

**Figure 5 toxics-08-00041-f005:**
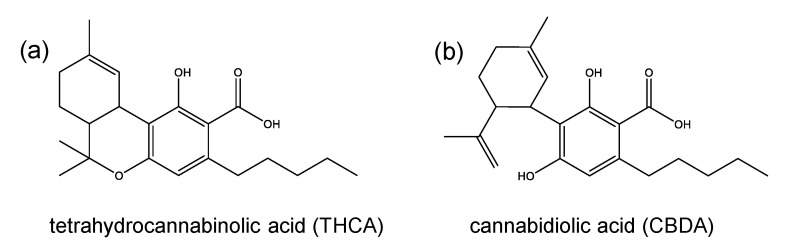
Chemical structures of (**a**) cannabidiolic acid (CBDA) and (**b**) Δ^9^-tetrahydrocannabinolic acid (Δ^9^-THCA).

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
