# Peer review of "Conversion of Cannabidiol (CBD) into Psychotropic Cannabinoids Including Tetrahydrocannabinol (THC): A Controversy in the Scientific Literature"

_toxics, 2020, doi:10.3390/toxics8020041_

Round 1

Reviewer 1 Report

* It would be advisable to include the identification of the CBD molecule in figure 3.

* It is recommended to do a general revision of the use of English (trying not to use "yet" so often).

* It would be advisable to rewrite the same information in another way to facilitate reading and understanding. For example:

- between lines 386-387 the authors repeat the word "effects" four times.

- between lines 492-493 the authors use “products” four times.

- between lines 518-519 “CBD” is used three times.

* There are some typographical errors, a detailed revision of the text is necessary. Some examples:

- Line 320 and 470; "conversation" should be replaced by "conversion"

- Line 335 and 498 some punctuation marks, such as some commas, are required before "which"; or after “CBD” in line 487.

- Line 447. ‘the decomposition of CBD in various solvents was controversially discussed between Turner et al. [94] who reported no decomposition and Fairbairn et al. [95] as well as Parker et al. [96] who indeed found CBD to be decomposed in different solvents under storage conditions.’ à  Could the authors have intended to write products instead of solvents at the beginning of the paragraph?

* In general, it is recommended to delete the information provided from lines 279 to 291, the reasons are:

-The authors assert that a conversion from CBD to delta9-THC has been proposed by Garcia et al. However, in reference 59 it was only demonstrated the mechanism in the conversion of CBC to THC using the thermal isomerization as an efficient synthetic tool for interconverting cannabinoid analogues. This is a major mistake that must be rectified.

-In line 283 the theoretical thermal degradation from CBD to A9-THC was not indicated in any of the referenced articles [20, 37, 38]. This is a major mistake that must be rectified.

-“Yet, as the derivatization process was sometimes found to be not quantitative [57,60], even small amounts of CBD, which evaded derivatization may account for positive findings of A9-THC due to thermal reactions in the GC system. “. This information has not been shown in any of the two indicated references. This is a major mistake that must be rectified.

Author Response

It would be advisable to include the identification of the CBD molecule in figure 3.

Reply: As suggested by the reviewer, we added the labeling “CBD” to the respective chemical structure in Figure 3 (now Figure 4 due to changes made throughout the revision process)

It is recommended to do a general revision of the use of English (trying not to use "yet" so often).

Reply: As also suggested by the editor, the language of the whole manuscript was carefully revised. In this context, the word “yet” was removed in several incidences.

It would be advisable to rewrite the same information in another way to facilitate reading and understanding. For example: Between lines 386-387 the authors repeat the word "effects" four times.

Reply: We would like to stress, that scientific correctness is superior to readability and repeated words (as also criticized in the next points of the reviewer) are sometimes necessary to unambiguously report on results. Nevertheless, this section was carefully reviewed in regard to avoidance of the word “effects” and we removed the word “effects” in two out of the four incidences.

Between lines 492-493 the authors use “products” four times.

Reply: As requested by the reviewer, the word “product” was replaced by “goods” and “articles” in two incidences.

Between lines 518-519 “CBD” is used three times.

Reply: The word “CBD” was removed one time in order to improve readability.

There are some typographical errors, a detailed revision of the text is necessary. Some examples: Line 320 and 470; "conversation" should be replaced by "conversion"

Reply: The word “conversation” was replaced by “conversion” as correctly pointed out by the reviewer.

Line 335 and 498 some punctuation marks, such as some commas, are required before "which"; or after “CBD” in line 487.

Reply: Thanks to the reviewer for this helpful comment. The commas were added as requested. Further the manuscript has been carefully checked for commas and they were added or removed in respective incidences.

Line 447. ‘the decomposition of CBD in various solvents was controversially discussed between Turner et al. [94] who reported no decomposition and Fairbairn et al. [95] as well as Parker et al. [96] who indeed found CBD to be decomposed in different solvents under storage conditions.’ à  Could the authors have intended to write products instead of solvents at the beginning of the paragraph?

Reply: Thanks to the reviewer for this kind comment. We double-checked all literatures. In fact, all publications cited here (Turner, Fairbairn and Parker) investigated the decomposition of CBD in various solvents. Therefore, this section is correct in its initial form and no changes were made.

In general, it is recommended to delete the information provided from lines 279 to 291, the reasons are: The authors assert that a conversion from CBD to delta9-THC has been proposed by Garcia et al. However, in reference 59 it was only demonstrated the mechanism in the conversion of CBC to THC using the thermal isomerization as an efficient synthetic tool for interconverting cannabinoid analogues. This is a major mistake that must be rectified.

Reply: We agree to the reviewer and admit, that CDB was mistaken for CBC in regard to this publication. We therefore carefully reviewed this paragraph but decided not to remove the whole section (as suggested by the reviewer). The reason for this decision is that the argument we present here is not founded on the study of Garcia et al. (which was only used as an example) but rather on the observation that most studies using GC/MS methods detected THC while such using LC/MS methods did not. Hence, we conclude that a possible thermal conversion of CBD to THC might be caused by GC/MS measurements. This is not in contrast or even interfering with Garcias report on CBC. However, our argument cannot be directly traced back to Garcia et al. as correctly pointed out by the reviewer. To clarify this, we added the sentence “This could most likely also apply to CBD and Δ9-THC.” to the manuscript.

In line 283 the theoretical thermal degradation from CBD to A9-THC was not indicated in any of the referenced articles [20, 37, 38]. This is a major mistake that must be rectified.

Reply: We agree with the reviewer, that thermal degradation was not reported in any of the cited literatures. However, this is not the point we tried to make here. In fact, we wished to highlight the observation that all studies that used GC/MS found THC, while most of the studies using LC/MS methods did not. The cited articles are examples for studies were GC measurements resulted in detection of THC. Neither did the authors of the cited articles, nor do we have scientific proofs for the thermal degradation, so we only report on the observation. To clarify that we do not present scientifically proven facts but rather summarize former results and speculate on their reasons, we already introduced our conclusion by the phrase “Hence, the question arises” in the initial submission. Therefore, we decided not to make changes to this section.

“Yet, as the derivatization process was sometimes found to be not quantitative [57,60], even small amounts of CBD, which evaded derivatization may account for positive findings of A9-THC due to thermal reactions in the GC system. “. This information has not been shown in any of the two indicated references. This is a major mistake that must be rectified.

Reply: We agree to the reviewer, that both literatures do not report on thermal derivatization of CBD to THC. They only report on the fact, that derivatization was found to be not quantitative. This is why the literature is not cited at the end of the sentence but at the respective part (i.e. where the sentence indicates that derivatization is not quantitative). In order to clearly highlight that the conclusion that THC may still be found due to conversion from small amounts of underivatized CDB is a conclusion derived by us, we slightly adjusted this section which now reads as follows:

“However, the derivatization process was sometimes found to be not quantitative [57,60]. Hence, it cannot be excluded that even small amounts of CBD, which evaded derivatization may account for positive findings of Δ9-THC due to thermal reactions in the GC system.”

Reviewer 2 Report

Toxics: Review for Manuscript No. Toxics-801304 entitled “Conversion of Cannabidiol (CBD) into Psychotropic Cannabinoids including Tetrahydrocannabinol (THC): A Controversy in the Scientific Literature”

Research on cannabidiol is intriguing under the circumstances that it has been widely used in food supplements and therapeutic drugs, in which Epidiolex was approved by FDA for the treatment of epilepsy disorders. The authors review the conversion of cannabidiol, which is known as a non-psychotropic compound, into psychotropic cannabinoids including tetrahydrocannabinol. Most of potential reasons for the presence of psychotropic cannabinoids in cannabidiol samples under in vitro and in vivo conditions were carefully discussed. The manuscript was well written, and its content is interesting to researchers in the field of cannabinoid research. I suggest this work be published in Toxics in its current manner.

Author Response

Reply: Thanks to the reviewer for this kind assessment of our paper!

Reviewer 3 Report

The review titled “Conversion of Cannabidiol (CBD) into Psychotropic 3 Cannabinoids including Tetrahydrocannabinol 4 (THC): A Controversy in the Scientific Literature”, by Golombek and colleagues sheds light on the potential reexamination of CBD as a non-psychotropic cannabinoid. Evidence presented in this paper suggests that CBD can be converted to psychotropic metabolites under certain conditions. The authors also rightfully highlight the paucity of animal and human studies documenting potential psychotropic effects of CBD metabolites. The idea for the paper is novel, and there are not a lot of reviews addressing this topic. The paper is also well-written. The authors did a great job on summarizing the results from the studies that they referenced here. The sections 3.3, 3.5 and 3.6 are especially very informative and interesting.  However, I do think this paper is missing some key points on CBD research. Additionally, in terms of structure, it is not very cohesive (lack of flow) and appears disjointed in parts.

Please address the following concerns:

  • I think this subject is rather contentious and the authors need to provide enough evidence on both sides of the argument. CBD has been well-documented to have multiple physiological effects with limited psychotropic effects. There needs to be a complete section on this. While the idea of potential conversion of CBD to psychotropic cannabinoids, resulting in neurological adverse effects, is novel, there is not enough evidence in animal/human studies to suggest that this is a concern. The authors rightfully point this out, but do not give enough credence to the overwhelming number of studies in animals and humans, that have shown limited or no psychotropic effects.  
  • The sections of the paper needs rearrangement for a better flow. Undoubtedly, the strongest and most important sections of the paper are sections 3.3, 3.5 and 3.6. “Psychotropicity of cannabinoids” (section 3.5), should be listed much earlier in the paper, that is before focusing on CBD studies. Also, this section should introduce all the conversion products succinctly, and in a clear manner. Additionally, make a distinction between metabolites that are psychotropic and not. Apart from this section, the section on analytical challenges (3.4), while very informative, seems misplaced. It needs to tie in better with the other sections. Can it be integrated with the other sections?
  • Please include a section on the pharmacokinetics of CBD. This should include all the well-known metabolites of CBD in vivo. Also, any drug/food/alcohol interactions with CBD in vivo, that could enhance its metabolism, could be listed here. Additionally, the potential impact of high doses on liver enzymes can be introduced here.
  • In the introduction section, the sentence “The observation that CBD products may still induce some psychotropic effects, despite CBD being known to be a non-psychotropic cannabinoid, recently again highlighted the importance of this question” is misleading. The publication referenced describes contamination of CBD with the psychoactive Δ9 -tetrahydrocannabinol (Lachenmeier et al., 2020). This publication does not highlight the possibility of psychotropic effects due to a conversion CBD to psychotropic cannabinoids. This needs to be clarified.
  • Possibly the strongest and the most clinically-relevant piece of evidence presented in favor of CBD’s conversion to psychotropic metabolites, is during improper storage. I feel this section needs to be highlighted, and discussed more in the conclusion and introduction sections.
  • Figure 3 is not well-presented, and very difficult to make sense of. Please present the conversions in an orderly fashion.
  • Is it necessary to have molecular weight and molecular formula in the table for the cannabinoids? Can be removed if it is not adding anything to the paper.
  • Most sections in the paper have several paragraphs. I find this unnecessary, and it breaks the reader’s flow. Please improve transitions and limit the number of paragraphs in each section to two or three.
  • Grammatical corrections are needed for the following:
    1. Page 1, Abstract: Cannabidiol (CBD) is a naturally occurring, non-psychotropic cannabinoid of the hemp plant Cannabis sativa L. CBD has been known to induce several physiological and pharmacological effects. While CBD is approved as a medicinal product subject to prescription, it is 16 also widely sold over the counter (OTC) in the form of food supplements, cosmetics and electronic 17 cigarette liquids.
    2. Section 3.3 and conclusion: Please change “human organism” to humans.
    3. Section 3.1: Please make changes.
      1. CBD is dissolved in dichloromethane
      2. CBD dissolved in ethanol
    4. The use of punctuation (for instance comma) is missing in several places. Please check this. One example is mentioned below
      1. Despite the increasing number of 254 publications driven by the ongoing scientific controversy, the question if conversion processes of 255 CBD may lead to psychotropic effects in the human body, is still not answered conclusively.

Author Response

The review titled “Conversion of Cannabidiol (CBD) into Psychotropic Cannabinoids including Tetrahydrocannabinol (THC): A Controversy in the Scientific Literature”, by Golombek and colleagues sheds light on the potential reexamination of CBD as a non-psychotropic cannabinoid. Evidence presented in this paper suggests that CBD can be converted to psychotropic metabolites under certain conditions. The authors also rightfully highlight the paucity of animal and human studies documenting potential psychotropic effects of CBD metabolites. The idea for the paper is novel, and there are not a lot of reviews addressing this topic. The paper is also well-written. The authors did a great job on summarizing the results from the studies that they referenced here. The sections 3.3, 3.5 and 3.6 are especially very informative and interesting. However, I do think this paper is missing some key points on CBD research. Additionally, in terms of structure, it is not very cohesive (lack of flow) and appears disjointed in parts.

Reply: Thanks to the reviewer for this helpful comment and guidance in improving the flow of the paper. We have conducted the requested revisions as further specified below.

Please address the following concerns: I think this subject is rather contentious and the authors need to provide enough evidence on both sides of the argument. CBD has been well-documented to have multiple physiological effects with limited psychotropic effects. There needs to be a complete section on this. While the idea of potential conversion of CBD to psychotropic cannabinoids, resulting in neurological adverse effects, is novel, there is not enough evidence in animal/human studies to suggest that this is a concern. The authors rightfully point this out, but do not give enough credence to the overwhelming number of studies in animals and humans, that have shown limited or no psychotropic effects. 

Reply: There are many excellent reviews available on the multiple physiological effects of CBD. As it appears to be impossible to cover all aspects of this growing research area in one article, we deliberately excluded this aspect from the current manuscript. In order to avoid presenting biased results, we added multiple references on research articles and reviews of health-promoting effects related to CBD into the introduction and clarified the search strategy in the materials and methods section.

The sections of the paper needs rearrangement for a better flow. Undoubtedly, the strongest and most important sections of the paper are sections 3.3, 3.5 and 3.6. “Psychotropicity of cannabinoids” (section 3.5), should be listed much earlier in the paper, that is before focusing on CBD studies.

Reply: The sections were moved in the sequence as suggested by the reviewer. Literature and cross referencing were carefully updated accordingly.

Also, this section should introduce all the conversion products succinctly, and in a clear manner.

Reply: We have added an introduction of the conversion products at the beginning of the new section 3.2. Indeed, it appeared to us even during the preparation of the initial submission that a detailed introduction of all conversion products in text form misses flow and is hard to read. Hence, we summarized all products in Table A1, which is also referred to in the corresponding section.

Additionally, make a distinction between metabolites that are psychotropic and not.

Reply: In terms of psychotropicity of the metabolites we refer to new chapter 3.1 and the Table in the Appendix, where a clear distinction between psychotropic and non-psychotropic metabolites is indicated wherever possible.

Apart from this section, the section on analytical challenges (3.4), while very informative, seems misplaced. It needs to tie in better with the other sections. Can it be integrated with the other sections?

Reply: The section about analytical challenges was moved to the beginning because the problems may arise in either thermal degradation studies, in vitro and in vivo studies. Therefore, it is actually more logical to introduce this at the beginning. Thanks to the reviewer for this kind suggestion.

Please include a section on the pharmacokinetics of CBD. This should include all the well-known metabolites of CBD in vivo. Also, any drug/food/alcohol interactions with CBD in vivo, that could enhance its metabolism, could be listed here. Additionally, the potential impact of high doses on liver enzymes can be introduced here.

Reply: These are new aspects outside the scope of our review, which would considerably expand the text, which is already very long. We strongly believe that this is a topic of major interest and a separate 6000-word review on pharmacokinetics, interactions, and metabolism could be written. However, we do not want to provide this is the context of the present review that is dedicated to CBD conversion. Hence, we decided not to add an additional section as kindly suggested by the reviewer.

In the introduction section, the sentence “The observation that CBD products may still induce some psychotropic effects, despite CBD being known to be a non-psychotropic cannabinoid, recently again highlighted the importance of this question” is misleading. The publication referenced describes contamination of CBD with the psychoactive Δ9 -tetrahydrocannabinol (Lachenmeier et al., 2020). This publication does not highlight the possibility of psychotropic effects due to a conversion CBD to psychotropic cannabinoids. This needs to be clarified.

Reply: The sentence was clarified as requested by the reviewer.

Possibly the strongest and the most clinically-relevant piece of evidence presented in favor of CBD’s conversion to psychotropic metabolites, is during improper storage. I feel this section needs to be highlighted, and discussed more in the conclusion and introduction sections.

Reply: Indeed, we share the reviewer’s opinion on the special importance of storage stability. However, we tried to keep the introduction as neutral as possible to avoid biasing the reader. Therefore, we decided to further stress the aspect on storage stability, which is carefully reviewed in this manuscript in the conclusion part of the manuscript.

Figure 3 is not well-presented, and very difficult to make sense of. Please present the conversions in an orderly fashion.

Reply: Actually, we like our Figure 3 (now Figure 4 due to changes throughout the revision process) as it presents a concise overview of possible reactions. For a presentation in an orderly fashion we kindly refer to the detailed table in Appendix A.

Is it necessary to have molecular weight and molecular formula in the table for the cannabinoids? Can be removed if it is not adding anything to the paper.

Reply: Thanks to the reviewer for this kind notification. Indeed, this does not add anything to the paper and was therefore deleted as requested.

Most sections in the paper have several paragraphs. I find this unnecessary, and it breaks the reader’s flow. Please improve transitions and limit the number of paragraphs in each section to two or three.

Reply: Actually, we believe paragraphs increase readability, but nevertheless merged very short paragraphs as requested by the reviewer.

Grammatical corrections are needed for the following: Page 1, Abstract: Cannabidiol (CBD) is a naturally occurring, non-psychotropic cannabinoid of the hemp plant Cannabis sativa L. CBD has been known to induce several physiological and pharmacological effects. While CBD is approved as a medicinal product subject to prescription, it is also widely sold over the counter (OTC) in the form of food supplements, cosmetics and electronic cigarette liquids.

Reply: Thanks to the reviewer for the helpful comment. The sentences were included in the manuscript as requested.

Section 3.3 and conclusion: Please change “human organism” to humans.

Reply: The word “human organism” was replaced by “humans” as suggested by the reviewer.

Section 3.1: Please make changes. CBD is dissolved in dichloromethane, CBD dissolved in ethanol

Reply: The word “solved” was replaced by “dissolved” as suggested by the reviewer.

The use of punctuation (for instance comma) is missing in several places. Please check this. One example is mentioned below: Despite the increasing number of publications driven by the ongoing scientific controversy, the question if conversion processes of CBD may lead to psychotropic effects in the human body, is still not answered conclusively.

Reply: A comma was added in the sentence as requested. Further, the manuscript was carefully checked for punctuation as also requested by Reviewer 1.

Round 2

Reviewer 1 Report

Thank you for so carefully clarifying all the points indicated.